# A Leucine-Rich Repeat Receptor-like Kinase TaBIR1 Contributes to Wheat Resistance against *Puccinia striiformis* f. sp. *tritici*

**DOI:** 10.3390/ijms24076438

**Published:** 2023-03-29

**Authors:** Yingchao Sun, Xiaojie Wang, Feiyang Liu, Haoyu Guo, Jianfeng Wang, Zetong Wei, Zhensheng Kang, Chunlei Tang

**Affiliations:** State Key Laboratory of Crop Stress Biology for Arid Areas and College of Plant Protection, Northwest A&F University, Xianyang 712100, China; sunyc316@nwafu.edu.cn (Y.S.); wangxiaojie@nwsuaf.edu.cn (X.W.); liufeiyang@nwafu.edu.cn (F.L.); ghy2611625@nwafu.edu.cn (H.G.); ipp@nwsuaf.edu.cn (J.W.); wzt132@nwafu.edu.cn (Z.W.)

**Keywords:** cell surface receptor-like kinases (RLKs), stripe rust, immune responses, co-receptor

## Abstract

Plant cell surface-localized receptor-like kinases (RLKs) recognize invading pathogens and transduce the immune signals inside host cells, subsequently triggering immune responses to fight off pathogen invasion. Nonetheless, our understanding of the role of RLKs in wheat resistance to the biotrophic fungus *Puccinia striiformis* f. sp. *tritici* (*Pst*) remains limited. During the differentially expressed genes in *Pst* infected wheat leaves, a Leucine-repeat receptor-like kinase (LRR-RLK) gene *TaBIR1* was significantly upregulated in the incompatible wheat-*Pst* interaction. qRT-PCR verified that *TaBIR1* is induced at the early infection stage of *Pst*. The transient expression of TaBIR1-GFP protein in *N. bentamiana* cells and wheat mesophyll protoplasts revealed its plasma membrane location. The knockdown of *TaBIR1* expression by VIGS (virus induced gene silencing) declined wheat resistance to stripe rust, resulting in reduced reactive oxygen species (ROS) production, callose deposition, and transcripts of pathogenesis-related genes *TaPR1* and *TaPR2*, along with increased *Pst* infection area. Ectopic overexpression of *TaBIR1* in *N. benthamiana* triggered constitutive immune responses with significant cell death, callose accumulation, and ROS production. Moreover, *TaBIR1* triggered immunity is dependent on *NbBAK1*, the silencing of which significantly attenuated the defense response triggered by *TaBIR1*. TaBIR1 interacted with the NbBAK1 homologues in wheat, co-receptor TaSERK2 and TaSERK5, the transient expression of which could restore the impaired defense due to *NbBAK1* silencing. Taken together, *TaBIR1* is a cell surface RLK that contributes to wheat stripe rust resistance, probably as a positive regulator of plant immunity in a *BAK1*-dependent manner.

## 1. Introduction

Plants deploy cell surface-localized pattern recognition receptors (PRRs) for perceiving conserved molecules, such as pathogen/microbe-associated molecular patterns (PAMPs/MAMPs) or damage-associated molecular patterns (DAMPs) to trigger PAMP-triggered immunity (PTI) [1,2]. As the first layer of plant immunity, PTI is critical for plants to survive under the threat of numerous pathogenic microbes. The second layer is effector-triggered immunity (ETI), which is specifically dependent on the direct or indirect recognition of avirulence (Avr) effectors by cytoplasmic immune receptors, mainly the nucleotide-binding site (NBS) and leucine-rich repeat (LRR) class [3,4]. PTI defenses include the rapid ion fluxes, burst of reactive oxygen species (ROS), deposition of callose, activation of mitogen-activated protein kinases (MAPKs), and the expression of immune-related genes, while ETI events generally leads to robust hypersensitive response (HR), characterized by rapid and local cell death at the infection sites [4,5]. Recent studies demonstrated that some PRRs in PTI are also required for ETI, and the two immune signal pathways are potentiated and interconnected [6,7].

An increasing number of plant cell-surface receptors have been identified, all of which are reported to encode receptor-like kinases (RLKs) or receptor-like proteins (RLPs) [8]. RLKs contain a ligand-binding ectodomain, a transmembrane domain and an intracellular kinase domain, while RLPs lack the intracellular kinase domain. The ectodomain of them are quite divergent, thus allowing RLKs respond to different signals. Leucine-rich repeats (LRR)-containing receptor-like kinases (LRR-RLKs) represent the largest group of RLKs, with at least 223 members in Arabidopsis [9]. The two typical LRR-RLK in Arabidopsis FLAGELLIN SENSING 2 (FLS2) and EFR (Elongation Factor-Tu Receptor) recognize bacteria flg22 and the short, conserved N-terminal of the most abundant bacterial protein elongation factor-Tu though their extracellular LRRs, respectively [10,11,12]. The recognition of flg22 by FLS2 activates the intracellular kinase BIK1, which in turn causes phosphorylation of the NADPH oxidase RBOHD on plasma membrane and results in production of large amounts of ROS. Meanwhile, the intracellular PBL kinase are phosphorylated followed by activation of MEKK1-MKK1/2-MPK4 and MAPKKK3/5-MKK4/5-MPK3/6 signaling pathways [13,14,15,16]. The recognition of EFR and bacterial elf18 can activate the same downstream immune signal pathway as FLS2-flg22 [12]. Rice resistance gene *XA21*, encoding an LRR-RLK, recognizes the highly conserved Ax21 of *Xanthomonas oryzae* and confers broad spectrum resistance to the Gram-negative bacterium *Xanthomonas oryzae* pv. *oryzae* [17,18]. Although LRR-RLPs lack cytoplasmic domains, their function is similar to that of LRR-RLKs. The heterodimer formed by receptor protein CEBiP and CERK1 activates the downstream OsRLCK185 and MAPK cascades to mediate chitin signal transduction [19,20,21,22,23]. Generally, to date numerous LRR-RLKs are identified as crucial components in regulating plant immune responses.

RLK/RLP is usually associated with co-receptors to form extracellular immune receptor complex on the cell surface to transmit extracellular immune signal [1,8]. To date, almost all the identified PRRs rely on somatic embryogenesis receptor kinases (SERKs) or SOBIR1 [24,25]. SERKs directly complex with PAMP/MAMPs receptors upon ligand perception. For example, the Arabidopsis BAK1 (BRI1-ASSOCIATED KINASE 1), also known as SERK3, interacts with FLS2 or EFR upon flg22 or elf18 ligand perception [11,26]. BIR1 and BIR2 (BAK1-interacting receptor-like kinase), interacts with BAK1 and negatively regulates PRR complex in immunity [27,28]. In the absence of stimuli, BIR2 inhibits BAK1 interaction with FLS2. Upon perception of flg22, BIR2 dissociates from BAK1, thus allowing BAK1-FLS2 formation to trigger immune responses [27,29]. Arabidopsis SERK4 functions redundantly with SERK3 in regulating cell death containment and immunity [30]. Rice BAK1 orthologous OsSERK2 forms a constitutive complex with XA21 that confers resistance to *Xanthomonas oryzae* pv. *oryzae*. OsSERK2 is also required for XA3-mediated immunity as well as rice FLS2 signaling [31]. In tomato, SlSERK3 acts as positive regulator and is required for *Ve1*-mediated resistance to fungal *Verticillium* [32]. Therefore, SERK receptors interact with a broad array of receptors to modulate the immune signaling which may suggest they remain at the center stage for studying the roles of RLK.

The existing evidence shows that the immune pathway mediated by extracellular receptors is conservative among plant species. However, in contrast to the well-studied PRRs in model plants Arabidopsis, the characterization of key extracellular receptors in the major food crop wheat is very limited, especially during wheat–*Puccinia striiformis* f. sp. *tritici* (*Pst*) interaction. *Pst* is an obligate biotrophic fungus, causing devastating stripe rust disease on wheat worldwide. There is a lack of systematic and in-depth research on the role and regulation mechanism of extracellular receptors on wheat resistance to stripe rust. As a widely used experimental plant, *Nicotiana benthamiana* is susceptible to a wide variety of pathogens and provides facile methods such as virus-induced gene silencing or transient protein expression, making it a cornerstone for studying innate immunity and defense signaling in host-pathogen research [33].

In the present study, we functionally characterized a cell-surface located wheat LRR-RLK gene *TaBIR1*, which is highly induced upon avirulent *Pst* infection. Knockdown of *TaBIR1* significantly impaired wheat resistance to stripe rust and transient expression of *TaBIR1* triggered constitutive defense response in *N. benthamiana* requiring *NbBAK1*. Moreover, we revealed that TaBIR1 interacts with TaSERK2 and TaSERK5, which could restore the deficiency of *NbBAK1* in triggering plant immunity. The present investigation aimed to identify a wheat LRR-RLK TaBIR1 that positively regulates wheat resistance to stripe rust, which probably associates with TaSERK2 and TaSERK5 to confer immunity. 

## 2. Results

### 2.1. TaBIR1 Encodes a Conserved LRR-X Family Serine/Threonine Kinase

Among the differentially expressed genes in *Pst* infected wheat leaves analyzed by cDNA-AFLP [34], a transcript derived fragment (PST_576-6) is significantly upregulated with 5-fold in *Pst*-wheat interaction. Full-length cDNA cloning revealed an open reading frame (ORF) of 1,818 bp encoding a receptor-like kinase (RLK), which contains a signal peptide, extracellular LRRs (leucine rich repeats), a transmembrane region and a cytoplasmic kinase domain (Appendix A). Multiple alignments showed that this RLK displays conserved kinase subdomains across its homologues in other monocot and dicot plants, and shares 65.9% similarity to *BIR1* (AT5G48380) in Arabidopsis (Appendix A) that is classified into LRR-X subgroup. We thus named it *TaBIR1* (for *Triticum aestivum* BAK-Interacting Receptor-like kinase1). There are three homologues of *TaBIR1* located at chromosome 1A, 1B and 1D in wheat genome. The coding sequence of the three homologues of *TaBIR1* are 97.8% identical at the nucleotide level and have variations in 43 amino acids (Appendix A). Phylogenetic analyses revealed that *TaBIR1* is mostly close to BAJ86446.1 in *Hordeum vulgare* (Appendix A). 

qRT-PCR analyses showed that avirulent *Pst* CYR23 infection upregulated *TaBIR1* at the early infection stage such as 6 h post infection (hpi) and reaches the peak at 24 hpi, about 15-fold higher than that at 0 hpi (Figure 1). By contrast, the expression of *TaBIR1* were not obviously induced by the virulent *Pst* CYR31 infection (Figure 1). These results suggest that the expression of *TaBIR1* is induced in early avirulent *Pst* infection stages. 

### 2.2. TaBIR1 Localizes to the Plasma Membrane

TaBIR1 contains a predicted transmembrane domain, indicating its plasma membrane location. We examined its subcellular localization through transiently expressing green fluorescent protein (GFP) fused TaBIR1 by the cauliflower mosaic virus 35S promoter in *N. benthamiana*. The control cells expressing GFP alone showed a nuclear/cytoplasmic/plasma membrane distribution, while the fluorescence of TaBIR1-GFP fusion protein was confined to the cell surface (Figure 2A). To rule out the interference of plant cell wall, we next performed plasmolysis treatment. Green fluorescence was clearly visualized on the separated plasma membrane in cells expressing TaBIR1-GFP, suggesting the plasma membrane-localization of TaBIR1-GFP (Figure 2A). Immunoblotting analyses detected the corresponding bands of GFP and TaBIR1-GFP, indicating their successful expression (Figure 2B). To further confirm the plasma membrane localization of TaBIR1, we expressed TaBIR1-GFP in wheat mesophyll protoplasts. TaBIR1-GFP signal was observed on the plasma membrane of wheat cells (Figure 2C). Taken together, these data proved that TaBIR1 is localized on the plasma membrane.

### 2.3. Silencing of TaBIR1 Compromise Wheat Resistance against Pst

To investigate the function of *TaBIR1* in wheat resistance to stripe rust, we transiently knocked down the expression of *TaBIR1* in wheat using Barely Stripe Mosaic Virus (BSMV) mediated gene silencing (VIGS). Twelve days after BSMV inoculation on the second leaves, the fourth leaves of BSMV-infected plants appeared chlorotic mosaic symptoms, and wheat plants inoculated with BSMV:*TaPDS* showed obvious photobleaching, indicating the efficacy of virus-induced gene silencing (Figure 3A). At sixteen days post-inoculation of avirulent *Pst* race CYR23, the fourth leaves of control plants inoculated with BSMV:*GFP* exhibited significant hypersensitive response (HR) without sporulation, while a few uredinospore pustules were formed on wheat leaves of *TaBIR1*-knockdown plants (Figure 3B). Silence efficiency analyses showed that *TaBIR1* was downregulated by 54.2%, 81.8% to 64.7% at 0, 24 and 48 hpi (Figure 3C). qPCR revealed that fungal biomass in *TaBIR1*-knockdown plants was increased by 55.1% than that in the control plants at 10 dpi (Figure 3D). Histological observation showed larger infection area of *Pst* in *TaBIR1* silenced leaves at 120 hpi compared to that in the control leaves (Figure 3E,F), suggesting the promoted *Pst* growth and development in *TaBIR1*-knockdown plants. These data demonstrated that silencing of *TaBIR1* compromised wheat resistance against *Pst*. 

To further characterize the decreased wheat resistance, we examined the host immune responses in *TaBIR1* silenced plants. DAB staining showed decreased H_2_O_2_ accumulation per infection site in *TaBIR1* silenced leaves than in control plants at 24 hpi (Figure 4A). Total ROS in *TaBIR1* silenced leaves was 57.6% less than that in control plants at 24 hpi (Figure 4B). In addition, the callose deposition was decreased by 69.4% in *TaBIR1* plants at 24 hpi (Figure 4C,D). The expression level of the defense related genes *TaPR1* and *TaPR2* were downregulated by 60.9% and 64.5%, respectively, in *TaBIR1* silenced plants (Figure 4E,F). The results together suggested that silencing of *TaBIR1* led to defective immune responses, indicating the positive role of *TaBIR1* in wheat resistance to stripe rust. 

### 2.4. Transient Over-Expression of TaBIR1 in N. benthamiana Triggered Defense Responses

*TaBIR1* homologue in Arabidopsis is well known in basal immunity modulation, we therefore analyzed the basal roles of *TaBIR1* in plant defense. Aniline blue staining showed that heterologous expression of *TaBIR1* in *N. benthamiana* induce significantly more abundant callose deposition accumulation comparing to expressing *GFP* alone (Figure 5A,B). Additionally, dramatically increased ROS content were detected in TaBIR1-GFP expressed leaves at 36 and 48 h compared to that in leaves expressing GFP alone (Figure 5C), and significant ROS production was observed by DAB staining in *N. benthamiana* leaves expressing TaBIR1-GFP (Figure 5D). Along with that, trypan blue staining assay showed extensive cell death occurred at 48 h after expressing TaBIR1-GFP (Figure 5E), even though it could not trigger a visible cell death. In addition, TaBIR1-GFP expression upregulated the expression of *NbPR1* and *NbPR2* (Figure 5F). These findings strongly suggested that *TaBIR1* could induce constitutive immune responses and cell death in *N. benthamiana.*

### 2.5. NbBAK1 Is Required for TaBIR1 Mediated Cell Death and Immune Responses in N. benthamiana

Growing number of LRR-containing RLKs (and RLPs) are reported to engage in plant defense dependent on co-receptors, mainly *BAK1* or related *SERK* (Somatic embryogenesis receptor-like kinase), and *SOBIR1* [24,35,36]. To test whether *TaBIR1* mediated responses in *N. benthamiana* was dependent on *BAK1* or *SOBIR1*, we transiently silenced *NbBAK1* and *NbSOBIR1* using TRV-mediated gene silencing. qRT-PCR analyses revealed that *NbSOBIR1* and *NbBAK1* expression were reduced by 88.9% and 59.6%, respectively, comparing to that in control leaves inoculated with TRV2:*GFP* (Figure 6A,B). In *NbBAK1* silenced leaves, transient expression of *TaBIR1* resulted in significantly reduced callose deposition compared to that in the control leaves, while no obvious change in *NbSOBIR1* silenced leaves (Figure 6C,D). *TaBIR1* triggered ROS accumulation and cell death were also clearly attenuated by *NbBAK1* silencing (Figure 6E,F), so as the expression of *NbPR1* and *NbPR2* (Figure 6G,H). These results suggested that *NbBAK1* is required for *TaBIR1* mediated immune responses in *N. benthamiana.*

### 2.6. TaSERK2 and TaSERK5 Interact with TaBIR1 and Could Restore Impaired Defense Due to NbBAK1 Silencing

There are multiple *SERK* genes and *SERK-like* genes in wheat. Previously five *SERK* homologous genes were cloned from the wheat genome sequences based on characteristic features of *SERKs* [37,38]. Here, we successfully obtained four of them in wheat variety Su11, named *TaSERK1*, *TaSERK2*, *TaSERK3*, *TaSERK5*. The four *TaSERKs* encode typical LRR-RLKs sharing similar protein domains (Appendix A). In *N. benthamiana* leaves co-expressing *TaBIR1*:cLUC with *TaSERK2*:nLUC or *TaSERK5*:nLUC, significant luciferase activity were detected, while no activity were observed in leaves co-expressing *TaBIR1*:cLUC and *TaSERK1*:nLUC, or *TaSERK3*:nLUC using split luciferase complementation (SLC) assay (Figure 7A). The results indicated that TaBIR1 could interact with TaSERK2 and TaSERK5, but not with TaSERK1 and TaSERK3 in *N. benthamiana.* To further verify the interaction, we performed co-immunoprecipitation (co-IP) assay in planta using intracellular domains of *TaBIR1* and *TaSERKs*. The corresponding proteins were successfully detected in inputs, and only TaSERK2 and TaSERK5 were co-immunoprecipitated by TaBIR1 (Figure 7B), further confirming the interaction between TaBIR1 with TaSERK2 and TaSERK5. 

Since *TaBIR1* mediated immune responses in *N. benthamiana* depending on *NbBAK1* and that TaBIR1 associates with TaSERK2/TaSERK5, we tested if *TaSERK2* and *TaSERK5* could restore *TaBIR1* triggered immunity in *NbBAK1* silencing plants. Comparing to the seldom callose deposits in *NbBAK1* silencing plants expressing *TaBIR1* alone, the co-expression of *TaBIR1* and *TaSERK2* or *TaSERK5* in *NbBAK1* silenced leaves led to much more abundant callose deposition, but not when co-expressed with *TaSERK1* or *TaSERK3* (Figure 8). Additionally, co-expression of *TaBIR1* and *TaSERK2* or *TaSERK5* result in significant ROS burst and cell death in *NbBAK1* silenced leaves (Appendix A). Together, these results indicated that wheat *TaSERK2* and *TaSERK5* could compensate the role of *NbBAK1*.

## 3. Discussion

Planting resistant wheat cultivars is the most effective way to control wheat stripe rust. Nonetheless, *P. striiformis* continuously adapts to overcome host immunity, resulting in the ineffectiveness of disease-resistant varieties. Identifying novel resistant resources of durable resistance is of great importance for sustainable control of stripe rust. Extracellular receptor-mediated resistance tends to be broad-spectrum and durable due to the recognition of conserved microbial molecules, which provides an important genetic resource for the creation of broad-spectrum disease resistance materials. 

Transfer of EF-Tu receptor EFR, which is unique to cruciferous plants, into tomato, tobacco, rice, wheat and other plants, can improve the broad-spectrum resistance of genetically improved crops to bacterial diseases [39]. The rice bacterial blight resistance gene *XA21*, which recognizes the highly conserved AX21 in bacteria, confers broad-spectrum resistance to *Xanthomonas oryzae* [17,18]. The rice transcription factor gene, *OsAP2/ERF152*, is transferred, which results in the induction of callose deposition, cell death, and enhanced resistance against infection by *Xanthomonas oryzae* pv. *oryzae* [40]. Additionally, its expression in Arabidopsis transgenics activated the MPK3/6 and exhibited broad-spectrum resistance to *Pseudomonas syringae* pv. *tomato* DC3000 and *Rhizoctonia solani* AG1-IA [41]. The introduction of the *LecRK-V* gene from *Haynaldia villosa* into wheat variety YANGMAI158 mediates broad-spectrum resistance to powdery mildew [42]. Exploration of the regulation mechanism of PRR receptor and signal transduction may provide an important theoretical basis for the creation of broad-spectrum disease resistance crop materials. However, because of the large genome of common hexaploid wheat, the identification and functional study of wheat extracellular receptor genes are very limited. Here, we identified an important plasma-membrane receptor *TaBIR1* that positively modulates plant immunity and contributes wheat resistance to stripe rust. Knockdown of *TaBIR1* compromised wheat immunity against *Pst* with impaired ROS burst, callose accumulation and cell death occurrence. Heterologous overexpression of *TaBIR1* in *N. benthamiana* activated basal immunity with HR-like cell death, suggesting its potential use in improving wheat resistance. Deciphering the mechanism of *TaBIR1* mediated immunity is critical for its proper use in resistance engineering. 

*BAK1*-interacting receptor-like kinase (*BIR1*) is a negative regulator of *BAK1* and *SOBIR1*-mediated immune response. In Arabidopsis, loss function of *BIR1* leads to constitutive activation of cell death, enhanced accumulation of SA and H_2_O_2_, and induction of *PR* genes dependent on *BAK1*, *EDS1*, *PAD4*, and *SOBIR1* in the absence of pathogen infection [29]. The inactivation of *BIR1* triggers resistance to TRV [43]. Similarly, transient silencing *GmBIR1*, a *BIR1* homologue gene in soybean, also resulted in activated defense response and enhanced resistance to *Pseudomonas syringae* pv. *glycinea* (*Psg*) and Soybean mosaic virus (SMV) [44]. It seems that the function of *BIR1* homologues as a negative regulator of immunity is conserved among different plant species. However, our study revealed the positive role of *TaBIR1* in inducing cell death, callose deposition and ROS production in *N. benthamiana*, the knockdown of which in wheat attenuates resistance to stripe rust with decreased ROS and callose accumulation. *BIR1* homologues between dicot Arabidopsis and monocot wheat plants seems to play divergent roles in plant immunity regulation.

In fact, exploration on the relevance of *BIR1* regulation in infected plants revealed that *BIR1* overexpression causes morphological defects and limited TRV accumulation in *N. benthamiana* cells. Additionally, transgenic Arabidopsis with inducible *BIR1* overexpression stimulated an autoimmune response as observed in *bir1-1* without pathogen infection [43]. Both low and high expression of *BIR1* were concomitant to defense responses, assuming that *BIR1*-associated phenotypes are dose-dependent, and there is a threshold for the proper function of *BIR1*. It is not clear whether there is such a threshold for *TaBIR1*. Probably, the observed triggered defense is due to exceeds of the threshold, leading to the mis-regulation of plant immunity. Knockout mutants of *TaBIR1* are needed to demonstrate its role in cell death control and defense response regulation under pathogen free and pathogen infection environment in the future study.

*BIR1* belongs to BIR family, including four members (*BIR1*-*BIR4*). Evidence showed that BIRs have evolved partially redundant but also distinct functions. *BIR2* knock mutant also exhibits enhanced PAMP responses, but does not have the dwarf morphology, like *bir1-1*. Over-expression of *BIR1* does not affect flg22-induced immune responses, while the over-expression of *BIR2* attenuates PAMP-triggered responses [27,28]. BIR2 constitutively interacts with BAK1 in the absence of ligands, which upon ligand activation is released from the complex, and thus, BAK1 can associate with the ligand-bound receptor complex partners. In contrast to the other BIRs, BIR3 associates not only with BAK1 but also with all functional SERKs, thereby exhibiting a general molecular mechanism compared to the relatively specialized function of *BIR2*. The overexpression of *BIR3* leads to a strong swarf phenotype and significant blocking of BR responses, indicating its strong impact on both BR- and microbe-associated molecular pattern (MAMP)-induced responses, but its weak cell death control. It seems that although the BIR family proteins share similar sequence and domain structure, duplication events within the BIR subfamily of LRR-RLKs have evolved new molecular functions and mechanisms [45,46,47]. BIR1 is an active kinase [29]. Multiple alignments showed that *TaBIR1* exhibits mutations in conserved kinase motifs comparing to *BIR1*, which likely influence its kinase activity. Moreover, the cell death and defense response in *bir1-1* is mediated by both BAK1 and SOBIR1, but TaBIR1 triggered cell death in BAK1-dependent manner but not SOBIR1, suggesting their different downstream signaling pathway. Further studies on the kinase activity of TaBIR1 and its functional pathway will be necessary to deeply illustrate its distinct functional manner. 

BAK1 is a central regulator of innate immunity in multiple receptor complexes and interacts with *BIR1* family member [29,42,48]. TaBIR1 affects plant cell death and defense response required BAK1. Interestingly, we demonstrated that TaBIR1 associates with wheat BAK1 homologues TaSERK2 and TaSERK5. Moreover, TaSERK2 and TaSERK5 could compensate the repaired defense response caused by *BAK1* silencing. These results indicate the general regulatory role of BAK1 homologues in immune signaling by membrane-associated receptors among plant species as co-receptors, and a functional link of TaBIR1 with co-receptors. They may exist in the uncovered PRR complex since all of them are regulatory kinases that not response for ligand binding. BAK1, on one hand, could serve as a positive regulator in basal resistance against the bacterial pathogen *Pseudomonas syringae* pv*. tomato* DC3000 and the oomycete pathogen *Hyaloperonospora arabidopsidis*, and on the other hand, is involved in negative regulation of R protein mediated immunity. Knocking out both *BAK1* and its close homolog *BKK1* lead to strong autoimmune phenotypes [49]. In this way, the specifically regulatory roles of TaBIR1 are possibly determined by the involved receptor complex under different signaling pathway. The role of BAK1-resistance pathway needs further investigation to elucidate its contribution to TaBIR1 mediated stripe rust resistance. Additionally, the additional components in TaBIR1-TaSERK2/TaSERK5 module, especially the PRR with its potential ligand needs to be further investigated. 

In summary, our work demonstrates a wheat extracellular receptor gene *TaBIR1*, which contributes to wheat resistance against *Pst*. We demonstrate that the membrane-localized receptor TaBIR1 functionally associated with TaSERK2/TaSERK5 to activate immune responses. The new insights gained from the characterization of this receptor-like kinase will help to engineer PRR-based broad-spectrum resistant crop varieties.

## 4. Materials and Methods

### 4.1. Plant Growth, Conditions and Pathogen Infection

Plant materials include wheat cultivar Suwon11 (Su11) and *N. benthamiana* were used in this study. For wheat seedlings, plants were grown in a controlled growth chamber (16 h:8 h, light:dark, 16 °C:10 °C). Fresh urediospores of *Pst* race CYR23 (avirulent on Su11), CYR31 (virulent on Su11), were collected and inoculated according to the procedures described previously [50]. As for *N. benthamiana*, seedlings were grown in the growth chamber at 23 °C (16 h:8 h, light:dark). *Agrobacterium tumefaciens* strain GV3101 was used in infiltration assay for transient expression in *N. benthamiana*.

### 4.2. Total RNA Extraction and qRT-PCR Analysis

For RNA isolation, samples of wheat and *N. benthamiana* leaves were collected and total RNA was isolated using Trizol reagent (Invitrogen, Carlsbad, CA, USA) according to the manufacturer’s instructions. Two microgram of total RNA was used for reverse transcription using Revert Aid First Strand cDNA Synthesis Kit (Ferments, Waltham, MA, USA). All qRT-PCR reactions were conducted on a CFX Connect Real-Time System (Bio-Rad, Hercules, CA, USA). Gene relative expression was analyzed using the comparative 2^−ΔΔCT^ method with three biological replications [51]. Data significance was evaluated by Student’s *t*-test. All primers used in qRT-PCR were listed in Appendix A.

### 4.3. Sequence Analysis, Alignment and Polymorphism Analysis

Prediction of conserved domains and motifs were performed by PROSITE Scan (http://prosite.expasy.org/scanprosite/, accessed on 8 May 2020) and SMART (http://smart.embl.de/, accessed on 8 May 2020). DNAMAN 6.0 was used to achieve the multi-sequence alignments. Phylogenetic tree was created by the MEGA software (Version 7.0, Kumar S, Philadelphia, PA, USA).

### 4.4. Subcellular Localization

For subcellular localization in wheat mesophyll protoplasts, seedlings were kept dark for one day just before use to block the chloroplast auto-fluorescence. Wheat mesophyll protoplasts were prepared using seven-day-old leaves with digestion in the enzyme buffer. Constructions of *TaBIR1-GFP* and *GFP* were transformed to protoplasts using polyethylene glycol (PEG) calcium transfection according to the method described by Yoo et al. [52]. Green fluorescence signals were monitored 18 h after transformation.

For subcellular localization in *N. benthamiana* cells, *Agrobacterium* GV3101 strain carrying *TaBIR1-GFP* and *GFP* reconstructed plasmids were cultured in LB (Luria-Bertani) medium containing 50 μg mL^−1^ rifampicin and 50 μg mL^−1^ kanamycin at 28 °C. Cells were collected after shaking and resuspended in acetosyringone (AS) buffer (10 mM MgCl_2_, 10 mM MES, pH 5.6, and 120 μM acetosyringone) at a final OD_600_ of 0.6 and then infiltrated into *N. benthamiana* leaves. GFP fluorescence was observed 48 h later. Plasmolysis was performed by submerging leaves into 0.8 M mannitol for 10 min. 

All photos were taken by FV1000 MPE confocal laser microscope (Olympus, Tokyo, Japan) with assays repeated at least three times. 

### 4.5. BSMV-Mediated Gene Silencing 

For Barley Stripe Mosaic Virus (BSMV) assay in wheat, two silencing fragments (VIGS-1, VIGS-2) were designed and combined to specifically knock down *TaBIR1* transcript. Each fragment was evaluated by BLASTn against wheat genome to ensure specific silencing. Both fragments were cloned and inserted into BSMV:γ vector as previously described [53]. Plasmids were linearized and transcribed using the T7 in vitro transcription kit (Promega) following the manufacturer’s instructions. The transcripts of BSMV:*TaBIR1* (containing VIGS-1, VIGS-2) mixed with BSMV:α, BSMV:β, and FES buffer were inoculated on the second leaves of two-leaf stage wheat seedlings. TaPDS encoding *T. aestivum* phytoene desaturase was used as silencing index. A fragment of *GFP* was used as negative control. After virus inoculation, infected plants were maintained in a controlled growth chamber at 25 °C. Twelve days later, the fully expanded fourth leaves were further inoculated with fresh *Pst* CYR23 urediospores and maintained at 16 °C. The *Pst* inoculated leaves were collected at 0, 24, 48 hpi for silencing efficiency and histochemical assays. Samples at 240 h for fungal biomass was analyzed as described [54].

### 4.6. TRV-Mediated Silencing in N. benthamiana

For silencing assay in *N. benthamiana*, *A. tumefaciens* GV3101 cells carrying pTRV2-*NbSOBIR1*, pTRV2-*NbBAK1*, pTRV2-*NbPDS*, pTRV2-*GFP* were mixed individually with *A. tumefaciens* harboring pTRV1 in 1:1 ratio to a final OD_600_ of 0.2 and then infiltrated into cotyledons of four-leaf-stage seedlings. The pTRV2-*NbPDS* was used as an index for effective silencing and pTRV2-*GFP* was used as negative control. Two weeks later, when pTRV2-*NbPDS* silencing plants showed photo-bleaching, GV3101 carrying recombinant HA tagged *TaBIR1* were infiltrated into the corresponding leaves of pTRV2-*NbSOBIR1*, pTRV2-*NbBAK1*, pTRV2-*NbPDS*, and pTRV2-*GFP* silencing plants. Samples were collected at 48 h for subsequent immune analysis.

### 4.7. Transient Over-Expression in N. benthamiana

For *A. tumefaciens*-mediated transient expression, GV3101 carrying the recombinant constructs of *GFP* fused full length *TaBIR1* and *GFP* control were cultured and infiltrated into 4-week-old *N. benthamiana* leaves individually using 1-mL needleless syringe. At 48 h later, GFP fluorescence was checked for successful expression and samples were collected for immune responses detection. 

### 4.8. Histochemical Observations and Immune Responses Detection

In the in situ detection of H_2_O_2_ in wheat challenged by *Pst*, leaves were incubated in the 1 mg mL^−1^ 3,3-diaminobenzidine (DAB) solution for 6 h under a strong light. After the DAB solution was absorbed, samples were destained (absolute ethyl alcohol: acetic acid, 1:1, *v*/*v*) until the chlorophyll was completely removed. Then, leaf segments were immersed in chloral hydrate overnight for more transparent. Samples were preserved in 50% glycerol and the DAB polymerized reddish–brown spots produced at infection site was observed in bright channel under Olympus BX-53 microscope (Olympus Corp., Tokyo, Japan). For observation of callose deposits, destained wheat leaves were stained with 0.05% aniline blue in 0.067 M K_2_HPO_4_ (pH 9.0) overnight [55] and observed under UV channel of Olympus BX-53 microscope. Number of callose deposits per mm^2^ were calculated by ImageJ software (1.52a version). For the observation of hyphae, the infected tissues were cleared with ethanol as described above and boiled in 2 mL of 1 M KOH at 100 °C for 10 min. Leaves were then washed three times with 50 mM Tris-HCl (pH 7.4) for 10 min, followed by wheat germ agglutinin (WGA) staining that conjugated to Alexa-488 (Invitrogen) and observed by an Olympus BX-53 microscope. For each sample, thirty infection sites that a substomatal vesicle formed were examined and calculated using DP-BSW software (Olympus Corp., Tokyo, Japan) with Student’s *t*-test for the statistical analyses.

In *N. benthamiana*, for H_2_O_2_ observation, whole leaves were immersed into 1 mg mL^−1^ DAB solution with gently shaking for 6 h under light and then boiled in absolute ethyl alcohol for destaining. Observation of callose deposits was performed following the same method in wheat. As for cell death assay, samples were collected and stained with Trypan Blue solution (10 mL of lactic acid, 10 mL of glycerol, 10 g of phenol, 10 mg of trypan blue, dissolved in 10 mL of distilled water) and boiled for 5 min, then decolorized in chloral hydrate until tissue completely transparent [56]. Leaves were observed under UV channel of Olympus BX-53 microscope. 

For the measurement of total reactive oxygen species (ROS) content in wheat and *N. benthamiana* leaves, samples were ground in liquid nitrogen, and extracted in PBS solution (1 g mL^−1^) with 12,000 rpm centrifugation at 4 °C for 10 min. Supernatants were used to determine ROS concentration according to the instructions of Plant ROS ELISA kit (BioTSZ, E60016).

### 4.9. Protein Interaction

The luciferase complementation assay was performed in *N. benthamiana* leaves as previously described [57]. Briefly, *A. tumefaciens* GV3101 harboring the nLUC and cLUC constructs (all respective RLKs fused at the N-terminal to keep LUC fragments in cytoplasmic) were co-infiltrated into *N. benthamiana* leaves. Forty-eight hours later, leaves were taken and treated with 0.5 mM luciferin. The LUC activity was measured by the Lumazone Pylon 2048B (Princeton, NJ, USA).

For Co-IP assay, intracellular domain (ICD) of all *TaSERKs* in fusion with HA and FLAG tag (HA at C-terminal, FLAG at N-terminal) were amplified and inserted into the PICH86988 vector. *A. tumefaciens* GV3101 carrying individual *TaSERKs*-ICD-PICH vectors together with *TaBIR1*-ICD-pBinGFP were co-expressed in *N. benthamiana*. The infiltrated leaves were collected at 48 h later, ground in liquid nitrogen and extracted in a buffer (50 mM Tris-HCl, pH 7.5, 150 mM NaCl, 10% glycerol, 10 mM DTT, 1 mM NaF, 1 mM Na_2_MoO_4_·2H_2_O, 1 mM PMSF, 1% [*v*/*v*] P9599 Protease Inhibitor Cocktail, 1% [*v*/*v*] IGEPAL CA-630 [Sigma-Aldrich] (St. Louis, MO, USA)). After 12,000 rpm centrifugation at 4 °C for 10 min, each supernatant was incubated with 25 μL GFP-Trap agarose beads (Chromotek, gta-20) for 2 h followed by four times washing. The supernatant and total protein were immunoblotted by anti-HA antibody (Abbkine, A02045) and anti-GFP antibody (Abbkine, A02020). All primers used are listed in Appendix A.

## 5. Conclusions

The current research characterized a wheat LRR-RLK *TaBIR1* that functions as a positive regulator of wheat resistance against the biotrophic fungus *P. striiformis*. As a membrane-localized RLK, TaBIR1 requires the co-receptor NbBAK1 to modulate plant basal immunity in *N. benthamiana*. Moreover, wheat BAK1 homologues TaSERK2 and TaSERK5 could restore the function of NbBAK1 in TaBIR1 triggered immunity, indicating a TaBIR1-TaSERK2/TaSERK5 immune signaling pathway, which possibly contribute to wheat resistance to *Pst*. Given the short LRRs of TaBIR1, we assume that it might participate in a receptor complex to modulate immune outputs for establishing plant resistance. Our findings will provide a further understanding on the regulation of immune receptor complex in hexaploid wheat, and a potentially utilized genetic resource in crop improvement.

## Figures and Tables

**Figure 1 ijms-24-06438-f001:**
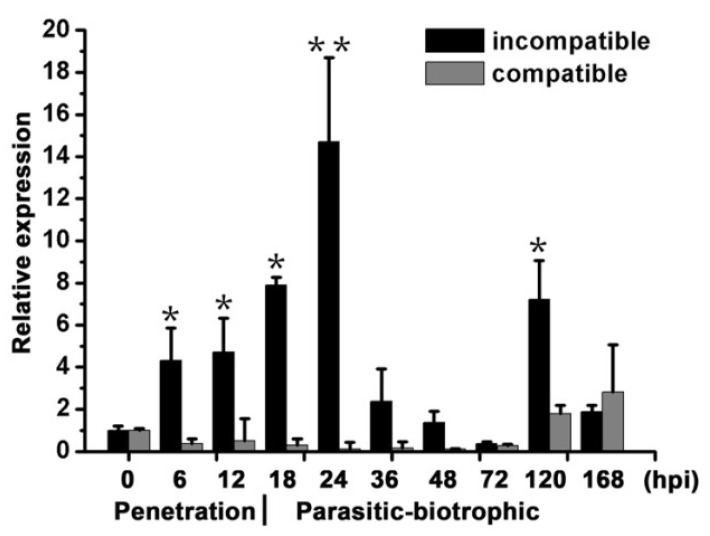
Expression profile of *TaBIR1* during wheat-*Pst* interaction. The second leaves of wheat cultivar Suwon11 were inoculated with fresh urediospores of avirulent *Pst* race CYR23 (incompatible interaction) or virulent *Pst* race CYR31 (compatible interaction) and sampled at different hours post-inoculation (hpi) for RNA extraction. Relative transcript levels of *TaBIR1* were calculated by the comparative threshold (2^−ΔΔCT^) method and normalized to the transcript level of *TaEF*. Values represent the means ± standard errors of three independent biological replicates. Asterisks indicate significant differences (* *p* < 0.05, ** *p* < 0.01) comparing to *TaBIR1* expression level at 0 hpi using Student’s *t*-tests.

**Figure 2 ijms-24-06438-f002:**
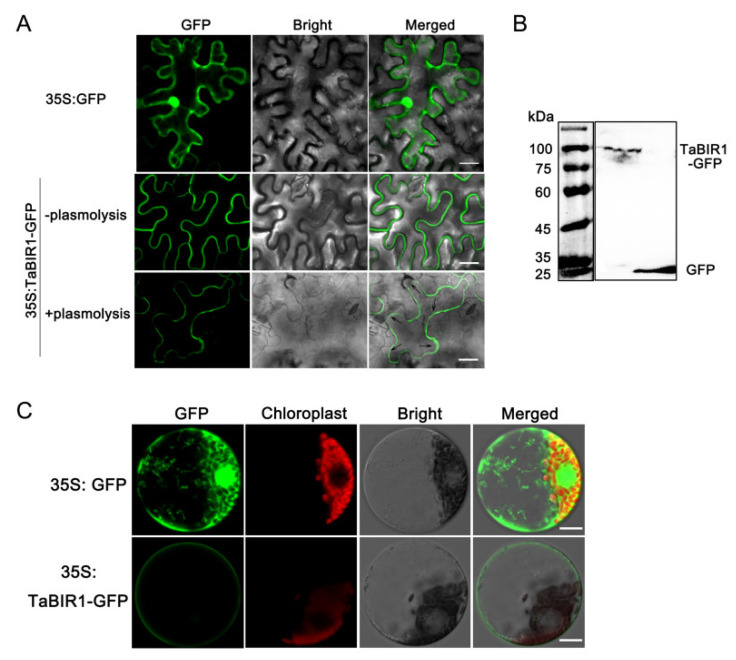
Subcellular localization of the TaBIR1-GFP. (**A**) Agrobacterium-mediated transient expression of GFP and TaBIR1-GFP driven by 35S promoter in *Nicotiana benthamiana* leaves. Fluorescence was observed under confocal microscopy at 48 h post infiltration. Arrows indicate the plasma membrane after plasmolysis treatment. Bars, 20 μm. (**B**) Protein Western blot of TaBIR1-GFP and GFP using anti-GFP polyclonal antibody. Protein ladder marker is shown at left. (**C**) Transient expression of GFP and TaBIR1-GFP under 35S promoter in wheat mesophyll cell protoplasts using PEG-mediated transformation. Images were taken under confocal microscopy. Bars, 20 μm.

**Figure 3 ijms-24-06438-f003:**
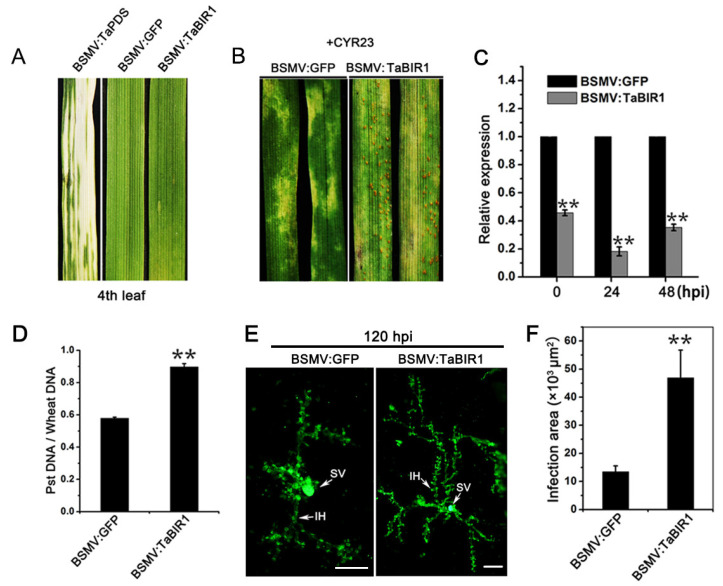
Transient silencing of *TaBIR1* via VIGS compromised wheat resistance to stripe rust. (**A**) Symptoms on the fourth leaves of wheat seedlings inoculated with BSMV:*TaPDS*, BSMV:*GFP* (negative control) and BSMV:*TaBIR1*. The second leaves of wheat cultivar Su11 were inoculated with recombinant BSMV and the virus phenotype were taken at 12 days post-inoculation (dpi). (**B**) Disease phenotypes of *TaBIR1*-silenced plants inoculated with *Pst* race CYR23 observed at 16 dpi. At 12 days post BSMV infection, the fourth leaves of wheat seedlings inoculated with BSMV:*GFP* and BSMV:*TaBIR1* were further challenged with avirulent *Pst* race CYR23. (**C**) Silencing efficiency of *TaBIR1* in *TaBIR1*-silenced plants at 0, 24 and 48 h post-inoculation (hpi) by qRT-PCR. (**D**) Relative *Pst* biomass in *TaBIR1*-silenced plants at 10 dpi. *PsEF* and *TaEF* were used as the internal reference genes. (**E**) Histological observation of *Pst* fungal development in *TaBIR1*-silenced plants at 120 hpi. SV, substomatal vesicle; IH, infection hyphae. Bars, 50 μm. (**F**) *Pst* infection area per infection site in *TaBIR1*-silenced plants at 120 hpi. In (**C**,**D**,**F**), asterisks indicate significant differences compared with controls using Student’s *t*-tests (** *p* < 0.01).

**Figure 4 ijms-24-06438-f004:**
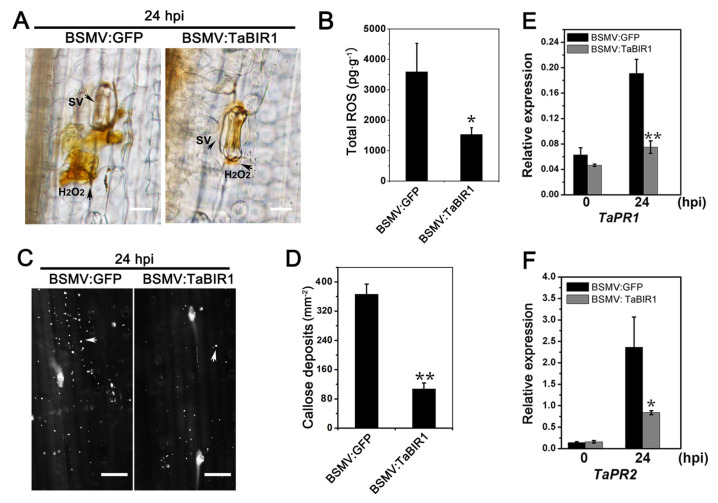
Impaired defense responses in *TaBIR1*-silenced plants. (**A**) Observation of DAB stained H_2_O_2_ in *Pst* infected *TaBIR1*-silenced plants at 24 hpi. Bars, 20 μm. (**B**) Determination of ROS production in *TaBIR1*-silenced plants. The fourth leaves of wheat seedings inoculated with BSMV:*GFP* and BSMV:*TaBIR1* were collected at 24 hpi for ROS content detection using HRP-labeled method. (**C**) Reduced callose deposition in *TaBIR1*-silenced plants. The fourth leaves of wheat seedings inoculated with BSMV:*GFP* and BSMV:*TaBIR1* were collected for aniline blue staining. White arrows shows the callose deposites. Bars, 50 μm. (**D**) Quantification of the number of callose deposits per mm^2^. Data are mean ± standard errors (n = 30) from three independent biological replicates. ** *p* < 0.01 (**E**,**F**) Expression profile of pathogenesis related genes *TaPR1* and *TaPR2* in *TaBIR1*-silenced plants using qRT-PCR. Wheat *TaEF-1α* was used as the internal reference gene. * *p* < 0.05, ** *p* < 0.01. Three biological replicates were performed.

**Figure 5 ijms-24-06438-f005:**
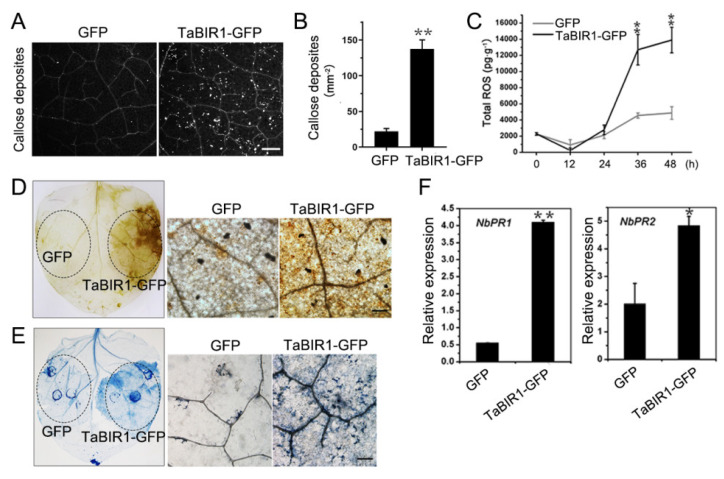
Transient expression of *TaBIR1* triggered immune responses in *Nicotiana benthamiana*. (**A**) Callose deposition stained by aniline blue in *N. benthamiana* leaves expressing GFP (the negative control) and TaBIR1-GFP. Bars, 200 μm. (**B**) Quantification of callose deposits per mm^2^. Values represent the means ± standard errors (n = 30). The experiment was repeated three times with similar results. ** *p* < 0.01 using two tails unpaired Student’s *t*-tests. (**C**) Determination of total ROS content in leave tissue at different time points after infiltration. (**D**) DAB staining of H_2_O_2_ production in *N. benthamiana* leaves expressing GFP and TaBIR1-GFP proteins. Scale bars, 100 μm. (**E**) Trypan blue staining of cell death in leaves expressing GFP and TaBIR1-GFP at 48 h after infiltration. Scale bars, 100 μm. (**F**) qRT-PCR analysis of *NbPR1* and *NbPR2* in *TaBIR1* expressing plant leaves. Values represent the means ± standard errors of three independent biological samples. Asterisks indicate significant differences compared with GFP control using Student’s *t*-tests. * *p* < 0.05, ** *p* < 0.01.

**Figure 6 ijms-24-06438-f006:**
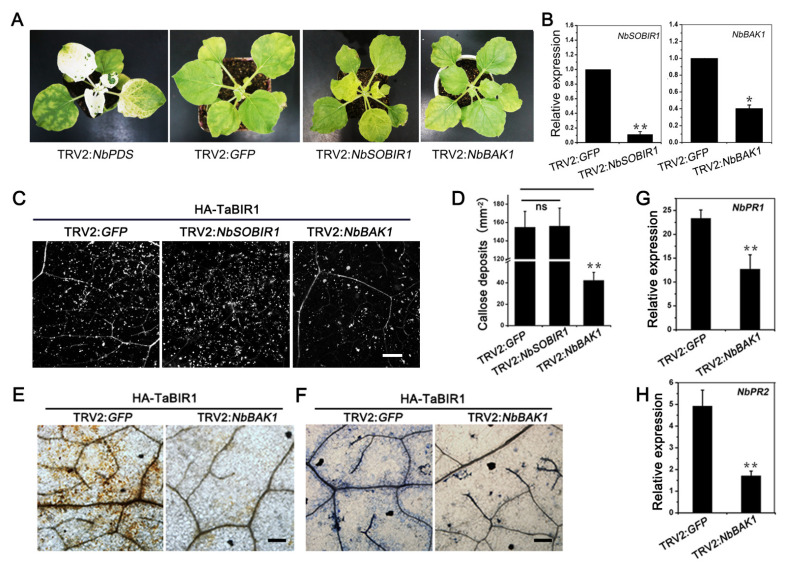
*NbBAK1* is required for *TaBIR1* triggered immunity. (**A**) Transient silencing of *NbSOBIR1* and *NbBAK1* in *N. benthamiana* mediated by tobacco rattle virus (TRV). Photobleaching was observed on plant leaves inoculated with TRV2:*PDS* two weeks later. GFP, negative control. (**B**) Silencing efficiency of *NbSOBIR1* and *NbBAK1* analyzed by qRT-PCR. Values represent the means ± standard errors using Student’s *t*-tests. (* *p* < 0.05, ** *p* < 0.01) (**C**,**D**) Transient silencing of *NbBAK1* reduced callose reposition triggered by *TaBIR1* in *N. benthamiana*. Plant leaves were inoculated with TRV2:*NbBAK1*, TRV2:*NbSOBIR1* and TRV2:*GFP*, on which two weeks later HA tagged TaBIR1 was transiently expressed via agrobacterium mediated infiltration. Forty-eight hours post infiltration, the leaves were stained by aniline blue for callose deposition observation. The number of callose deposits per mm^2^ was quantified. TRV2:*GFP* was used as the negative controls. Bars, 300 μm. ns, no significance, ** *p* < 0.01. Data are means ± standard errors from three biological replications. (**E**) Observation of DAB stained H_2_O_2_ and (**F**) trypan blue stained cell death in *GFP* and *NbBAK1* silenced leaves expressing HA-TaBIR1. Bars, 100 μm. (**G**,**H**) qRT-PCR analysis of *NbPR1* and *NbPR2* expression. Values represent the means ± standard errors. ** *p* < 0.01, significant differences from three biological replicates using two-tails unpaired Student’s *t*-tests.

**Figure 7 ijms-24-06438-f007:**
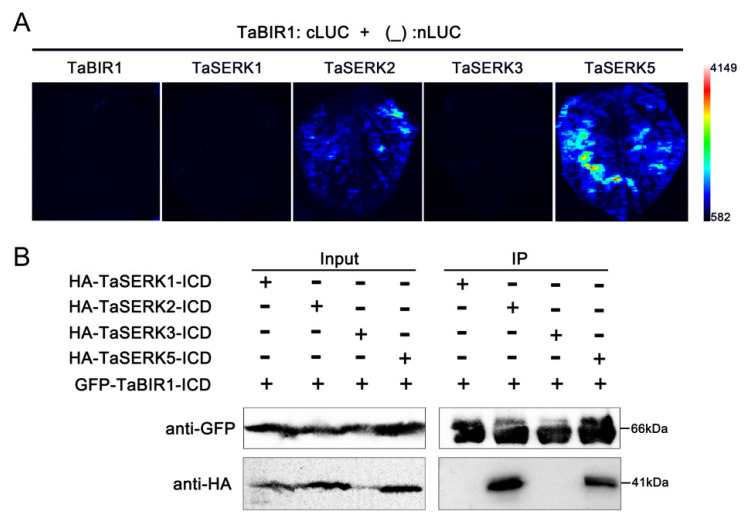
TaBIR1 associates with TaSERK2 and TaSERK5. (**A**) Interaction between TaBIR1 and TaSERK2, TaSERK5 confirmed by split-luciferase assay. *TaBIR1*:cLUC and indicated proteins with C-terminal nLUC were coexpressed in *N. benthamiana* by agroinfiltration. Images of chemiluminescence were recorded by applying 0.5 mM luciferin at 48 h after infiltration. (**B**) Interaction between TaBIR1 and TaSERK2, TaSERK5 verified by Co-IP assay. Intracellular domains (ICD) of *TaSERKs* were individually co-expressed with GFP tagged *TaBIR1* in *N. benthamiana.* Total proteins were extracted and immunoprecipitated by anti-GFP Agarose beads. The total proteins and immunoprecipitated proteins were detected using anti-GFP and anti-HA antibody.

**Figure 8 ijms-24-06438-f008:**
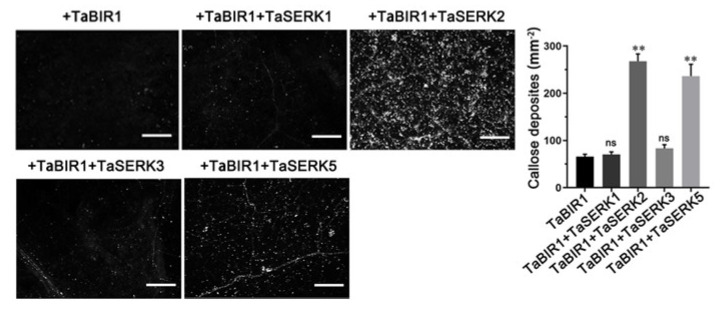
Heterologous expression of *TaSERK2* and *TaSERK5* restored the immunity triggered by *TaBIR1* in *NbBAK1*-silenced *N. benthamiana* leaves. Two weeks after inoculation of TRV2:*NbBAK1*, *TaSERKs* and *TaBIR1* were co-expressed in the corresponding plant leaves. Forty-eight hours later, the leaves were collected for aniline blue staining. The number of callose deposits per mm^2^ in *NbBAK1*-silenced leaves co-expressing *TaSERKs* and *TaBIR1* was quantified. Values represent the means ± standard errors from three independent biological samples (n = 30). Differences were assessed compared with that in leaves expressing TaBIR1 alone using Student’s *t*-tests (** *p* < 0.01, ns, no significance). Bars, 200 μm.

## Data Availability

The data used in this research is publicly available. The gene sequences can be found at https://www.ncbi.nlm.nih.gov/, accessed on 2 July 2021 (*TaSERK1*, Accession no. AK333001; *TaSERK2*, Accession no. AK333677.1; *TaSERK3*, Accession no. BT009223; *TaSERK5*, Accession no. BT009426). The data (results) presented in this research are available in the Appendix A.

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
