# Peer review of "A Leucine-Rich Repeat Receptor-like Kinase TaBIR1 Contributes to Wheat Resistance against Puccinia striiformis f. sp. tritici"

_ijms, 2023, doi:10.3390/ijms24076438_

Round 1
Reviewer 1 Report
ijms-2307906-peer-review-v1
The manuscript ‘A leucine-rich repeat receptor-like kinase TaBIR1 contributes to wheat resistance against Puccinia striiformis f. sp. tritici’ by Sun et al. is a well-conceptualized and well-written manuscript. I have a few points the author could address to improve the manuscript.
In the title, please mention ‘receptor like’ as ‘receptor-like’.
Please mention all the gene names in italics throughout the text (e.g., TaBIR1 in lines 17, 24, 25, 26….).
Please check the paragraph alignment of the abstract (Line 23-28). There is also an alignment issue in line no. 32-40 onwards. Please check.
The introduction is well-written, addressing the background/rationale and the importance of the work. The author may specify the aim of the work more clearly, and ‘Overall, here we identified’ (Line 101) may be rephrased as ‘The present investigation aimed to identify’. Please re-write sentences 101-103.
Please mention the need for N. benthamiana in the introduction section.
The results are well-presented.
Lile 107: ‘significantly upregulated’. Please mention the fold increase.
Line 145: ‘P < 0.05 / P < 0.01’ may be written in italics throughout the text.
Line 148: Please cross-check all the genes ‘TaBIR1’ mentioned in italics.
Line 177: ‘20 μ m’ may be written as ‘20 μm’ as mentioned in line 181.
Figure 3: The silencing efficiency was checked until 48 hpi. However, Pst fungal infection was checked for 120 hpi. Please justify..
Figure 6: Panel C could be zoomed out to 1000 µm for clear visibility of callose masses.
Line 395-396: The author may discuss 10.1007/s12284-009-9033-z
Line 492: Please change ‘wheat or N. benthamiana leaves’ to ‘wheat and N. benthamiana leaves'
Line 494: Please write ’Two microgram’ in letters at the beginning of the sentence.
Line 501-507: Please mention the date of access for the software used (Accessed on ….date….
Please mention the full form of GFP and PEG on their first appearance in the text.
Line 519: ‘FV1000 MPE confocal laser microscope (Olympus, Japan)’, please mention the city ‘Olympus, xxxx, Japan’.
Line 526: Please write ‘in vitro’ in italics.
Line 555: Please mention ‘In situ detection of H2O2’. Please use the unit ‘1mg/ml’ as ‘1mg ml-1’. Please check the de-staining agent ‘(absolute ethyl alcohol: acetic acid, 1:1, v/v)’ or ‘ethanol:acetic acid:glycerol (3:1:1)’
Please elaborate on the ‘In situ detection of H2O2’ method or cite a reference. The author may refer https://doi.org/10.1038/s41598-020-62317-z
Line 563: Please mention the version and date of access of ImageJ software used for counting the callose mass.
References may be arranged as per the journal pattern. Please cross-check the references cited with the list.
Although the overall grammar score is satisfactory, the authors are requested to polish the language once again.
The manuscript may be accepted with minor corrections.
Good luck with the revision.
Author Response
We would like to thank you for your careful reading, helpful comments, and constructive suggestions, which has significantly improved the presentation of our manuscript. We have carefully considered all comments from you and revised our manuscript accordingly. We summarize our responses to each comment and believe that our responses have well addressed all concerns from the reviewer. We hope our revised manuscript can be accepted for publication. Please see the attachment of Word "author to respond reviewer1-mdpi".

Reviewer 2 Report
In this manuscript, the authors identified a Leucine-repeat receptor like kinase in wheat, TaBIR1, involved in wheat stripe rust resistance. TaBIR1 is a plasma membrane localized protein and interacts with TaSERK2 and TaSERK5 to positively regulate plant immunity.
In my view, experiments are well designed, the evidence is solid/convincing, all the figures are clearly organized, and so the results are clearly shown. Overall, this manuscript is good enough to be published by IJMS.
Since TaBIR1 interacts with TaSERK2 and TaSERK5 while it does not interact with TaSERK1 and TaSERK3, I am just curious if silencing of TaSERK2 and TaSERK5, but not silencing TaSERK1 and TaSERK3, could compromise wheat resistance against stripe rust?
Only one minor revision point:
Page 11 line 451: "BIR1 is an active kinase."
This sentence should cite the following paper: "Gao, M.; Wang, X.; Wang, D.; Xu, F.; Ding, X.; Zhang, Z.; Bi, D.; Cheng, Y.; Chen, S.; Li, X.; et al. Regulation of cell death and innate immunity by two receptor-like kinases in Arabidopsis. Cell Host Microbe 2009, 6, 34–44."
Author Response
We would like to thank you for your careful reading, helpful comments, and constructive suggestions, which has significantly improved the presentation of our manuscript. We have carefully considered the comments from you and summarized our responses to each comment. We believe that our responses have well addressed the concerns from the reviewer. We hope our revised manuscript can be accepted for publication. Please see the attachment of Word "author to respond reviewer2-MDPI".
